# A VHF Band Small CRLH Antenna Using Double-Sided Meander Lines

Soyeong Lee [1] and Yong Bae Park [1,2,*]

1   Department of AI Convergence Network, Ajou University, Suwon 16499, Korea
2   Department of Electrical and Computer Engineering, Ajou University, Suwon 16499, Korea
*   Correspondence: yong@ajou.ac.kr; Tel.: +82-31-219-2358

**Abstract:** In this paper, a miniaturized very-high frequency (VHF) band antenna using both top and bottom meander lines is proposed. To design a compact size antenna in the VHF band, a Composite Right/Left-Handed (CRLH) transmission line is applied to antenna structure; additionally, both top and bottom meander lines were used to achieve a greater inductance. The CRLH transmission line unit cell operates at 88 MHz, and the fabricated antenna is designed by arranging 7-unit cells. The overall size of the proposed antenna is $0.087\lambda \times 0.02\lambda \times 0.0003\lambda$ at the lowest operating frequency, and the antenna operates at 84 MHz. The VSWR 3.5:1 reference operating bandwidth of the antenna is 2%. The received power of the proposed CRLH antenna was measured to verify the antenna performance.

**Keywords:** composite right/left-handed transmission lines; military antenna; VHF (very high-frequency); meander lines

## 1. Introduction

The antennas for military applications mainly use dipole, monopole, and whipped antennas operating in the HF/VHF/UHF bands, and require omnidirectional radiation patterns due to limited communication environments [1–3]. Since the physical size of the antenna is proportional to the wavelength, the antenna size inevitably increases as the frequency decreases. In fact, the length of monopole antenna in the VHF band is from 1 to 5 m, and there is a limit on the mobility and installation of communication equipment because an appropriate size of ground is required to have antenna characteristics. Therefore, it is essential to study the miniaturization of antennas used in military applications. There have been many studies to miniaturize the antenna, and, as one of the methods, an antenna miniaturization technique based on a CRLH (Composite Right/Left-Handed) TL (Transmission Line) is being studied [4–25].

The CRLH TL is one of the metamaterials and can be represented by combining the RH (Right-Handed) component, which is a parasitic component of the transmission line, and the LH (Left-Handed) component, which consists of a series capacitor and a parallel inductor. The CRLH TL antenna is designed by periodically arranging unit cells, and has two characteristics suitable for miniaturization of military antennas. First, the zeroth-order resonance of the CRLH TL has an omnidirectional radiation pattern suitable for military communication. Second, if the zeroth-order resonance of the CRLH TL is used, the physical size of the antenna is independent of the resonant frequency, so the antenna can be miniaturized [4]. The CRLH TL has different design factors affecting the resonant frequency depending on its open-ended and short-ended structure. In the case of an open-ended structure, parallel resonance occurs and the zeroth-order resonant frequency is determined by the parallel capacitor and parallel inductor. In the case of a short-ended structure, series resonance occurs and the zeroth-order resonant frequency is determined by the series capacitor and series inductor [5]. Since it is easier to control the parallel inductance than the series inductance on a plane, an open-ended structure is more often used to lower the resonance frequency of the antenna [5].

The CRLH TL antenna in previous studies implemented a series capacitor using the gap between patch and patch [8,9,12,16,19] or IDC (Interdigital Capacitor) [13,15,24,25]. A parallel inductor was implemented by connecting the via [8,9,13,16], spiral inductor [20,24,25], or meander line [6,9,13–15,19] located at the top of the antenna to the ground located at the bottom. When designing an open-ended antenna, most of the previous studies have been conducted in the UHF band because there is a limit to achieving a greater parallel inductance with the previous antenna structure [4–22]. There was also research on open-ended antenna using CRLH TL in the VHF band [23,24], but it is difficult to design an antenna operating at frequencies lower than 100 MHz using a spiral inductor and a meander line that exist only at the top. This is because the length of the inductor increases to operate at a low frequency, and, consequently, the physical size of the antenna increases. We need a miniaturized antenna structure for military applications operating in the VHF band. That is, a new CRLH TL unit cell structure having a greater parallel inductance without increasing the physical size is required.

In this paper, we propose a miniaturized antenna based on a CRLH transmission line in the VHF band using meander lines at both the top and bottom. The proposed antenna was miniaturized by using the zeroth-order resonant frequency of the CRLH transmission line, and the meander lines connected to the top and bottom were used to increase the parallel inductance. The unit cell consists of an interdigital capacitor and a meander line. In addition, the antenna is designed to increase the length of the meander line as the number of cells increases. The size of the unit cell is $0.01\lambda \times 0.02\lambda \times 0.0003\lambda$ ($37 \times 70.6 \times 1$ mm$^3$), and the size of the fabricated 7-unit cell antenna is $0.087\lambda \times 0.02\lambda \times 0.0003\lambda$ ($308.1 \times 70.6 \times 1$ mm$^3$) at 84 MHz. The proposed antenna was designed using an ANSYS high-frequency structure simulator (HFSS) and its performance was verified through simple measurement. The designs of the CRLH transmission line unit cell and antenna are described in Section 2. In Section 3, the simulated and measured results of antenna are introduced and compared. Finally, Section 4 concludes the paper.

## 2. Design of the CRLH Unit Cell and Antenna

Figure 1 shows the equivalent circuit of the CRLH TL [4]. The LH component consists of a series capacitor and a parallel inductor, and the RH component can be implemented with a series inductor and a parallel capacitor. The CRLH TL antenna was implemented by designing unit cells which combine the RH and LH components and which are periodically arranged. Although the miniaturized unit cells ($\ll \lambda/4$) are periodically arranged, the zeroth-order resonant frequency does not change significantly. Therefore, it was an important step to design the CRLH unit cell and determine the zeroth-order resonant frequency to design the antenna. As mentioned in the introduction, the antenna has an open-ended and a short-ended structure, and the zeroth-order resonant frequency can be calculated from the propagation constant of the unit cell [4]. In the open-ended structure, the zeroth-order resonant frequency is determined by the parallel element, as shown in Equation (1). $C_R$ is the capacitance formed between the top and bottom conductors and corresponds to a parasitic component. $L_L$ is implemented as a meander line or spiral inductor, and a greater parallel inductance is required to design an antenna operating at a low frequency. This can be achieved by increasing the length of the inductor or by using thinner meander lines.

$$\omega_{sh} = \frac{1}{\sqrt{L_L C_R}} \tag{1}$$

Figure 2a shows a CRLH TL unit cell structure with full ground at the bottom like a conventional CRLH TL unit cell, and Figure 2b shows a new unit cell structure proposed in this paper. The proposed unit cell was designed to operate at a low frequency by adding a meander line at the bottom in addition to the one at top to have a greater parallel inductance. The zeroth-order resonant frequencies of the two structures was confirmed using a dispersion diagram obtained by setting ports at both ends of a unit cell; moreover, the dispersion diagram was calculated from Equation (2) using the s-

parameter characteristics [16,17]. In Equation (2), *p* represents the period of the unit cell, and the frequency at which the propagation constant (β) becomes 0 is a zeroth-order resonant frequency.

$$\beta = cos^{-1}\left(\frac{1 - S_{11}S_{22} + S_{12}S_{21}}{2S_{21}}\right) \tag{2}$$

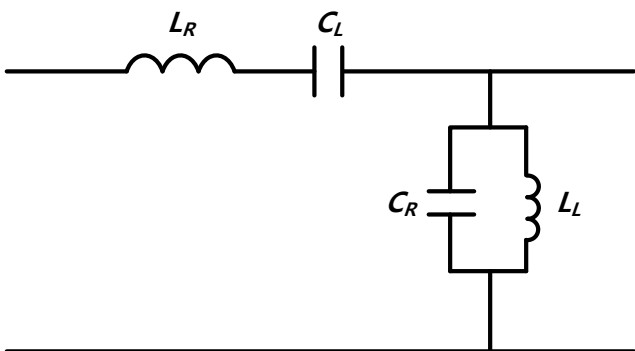

**Figure 1.** Equivalent circuit of CRLH transmission line.

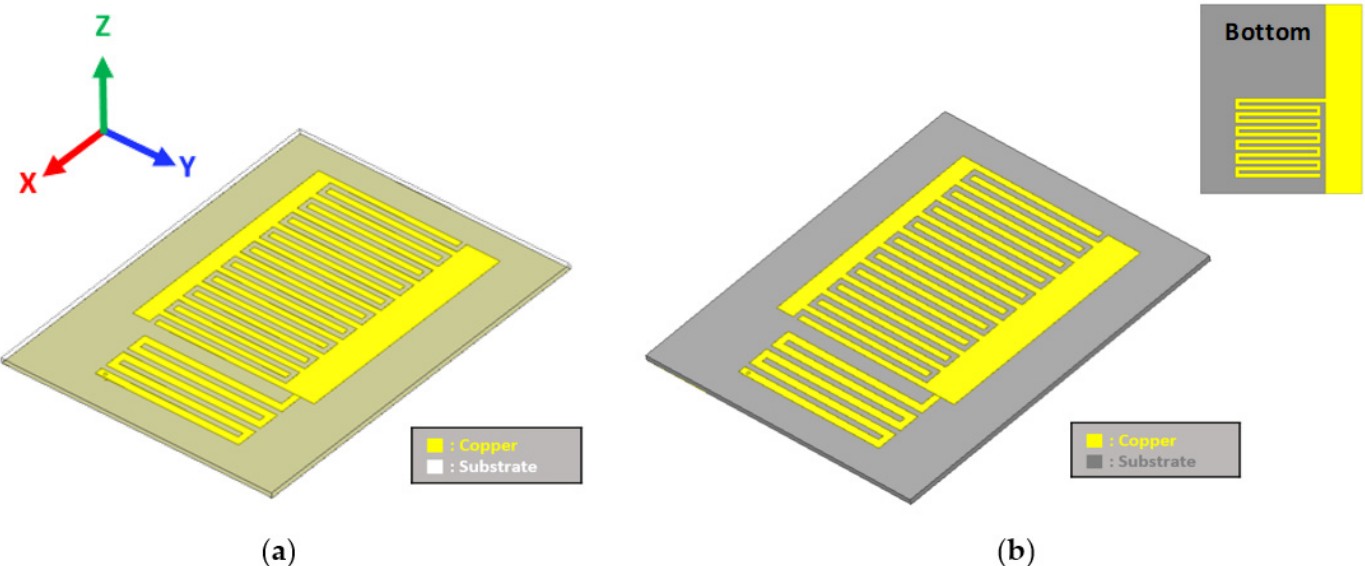

**Figure 2.** CRLH TL unit cell structure: (**a**) CRLH TL unit cell with full ground at the bottom; (**b**) CRLH TL unit cell with meander line at the bottom.

Figure 3a shows the dispersion diagram of Figure 2a, which confirms that the zeroth-order resonant of the unit cell is 131.6 MHz (β = 0 at 131.6 MHz). Figure 3b shows the dispersion diagram of Figure 2b, which shows that the zeroth-order resonant of the unit cell is 90.2 MHz (β = 0 at 90.2 MHz). In addition, the parallel element values of the two structures are shown in Table 1, and the parallel inductance is higher in the case where there is a meander line at the bottom of the unit cell than in the case where there is no meander line. A structure with a meander line only at the top also operates at low frequencies, but the length of the meander line must be increased to obtain a zeroth-order resonant frequency below 100 MHz. This means an increase in the physical size of the antenna. Moreover, it is difficult to increase parallel inductance simply by using thin lines because there is a manufacturing limit on planar antennas. Thus, the proposed unit cell is an advantageous structure for designing miniaturized antennas operating at low frequencies. The proposed CRLH unit cell used a Taconic TLY-5 ($\varepsilon_r = 2.2$) with a thickness of 40 mil. The unit cell size is $0.01\lambda \times 0.017\lambda \times 0.0003\lambda$ ($37 \times 58.65 \times 1$ mm³), and each

parameter is $f_s$ = 28.5 mm, $f_w$ = 1.2 mm, $ML_s$ = 1.5 mm, $ML_L$ = 32.25 mm, $via_r$ = 0.5 mm in Figure 4.

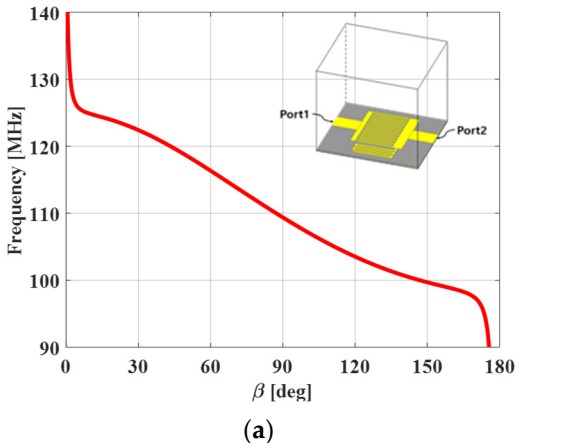

(**a**)

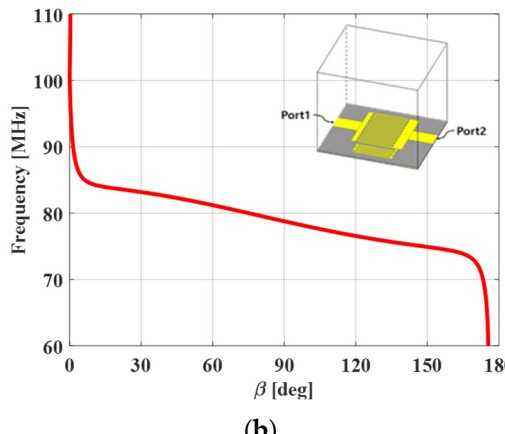

(**b**)

**Figure 3.** Dispersion diagram of the CRLH TL unit cell: (**a**) dispersion diagram of Figure 2a; (**b**) dispersion diagram of Figure 2b.

**Table 1.** Comparison of the parallel element values according to the CRLH TL unit cell structure.

|  | With Full Ground at Bottom (Figure 2a) | With Meander Line at Bottom (Figure 2b) |
|---|---|---|
| Size | $40.8 \times 58.65 \times 1.016$ mm$^3$ | |
| Resonant frequency | 131.6 MHz | 90.2 MHz |
| $C_R$ (Parallel Capacitance) | 20.6 pF | 19.45 pF |
| $L_L$ (Parallel Inductance) | 71 nH | 160 nH |

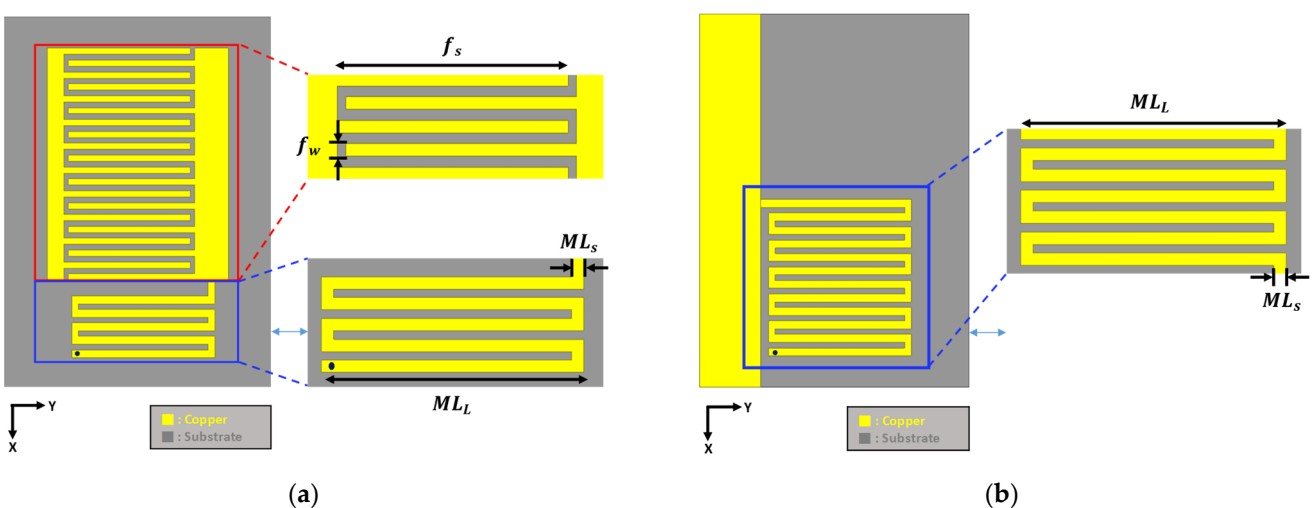

(**a**)

(**b**)

**Figure 4.** The unit cell of CRLH transmission line: (**a**) top view; (**b**) bottom view.

Figure 5a,b,c show the bottom of the antennas composed of 3-, 5-, and 7-unit cells, respectively. The top of the antenna is the same as that from Figure 2b. As can be seen in Figure 5, the proposed antenna was designed to increase the length of the meander line at the bottom as the unit cells increase. As the unit cell increases, the length of the meander line becomes longer than one unit cell, so the zeroth-order resonant frequency of

the proposed antenna moves to a lower side. The characteristics of each antenna according to the number of cells are shown in Table 2. In the case of the 3-cell antenna, the electrical size of the antenna has become very small. However, due to the low gain of the 3-cell antenna, it is not suitable for use in military applications. The 7-cell antenna has an overall size of $0.087\lambda \times 0.02\lambda \times 0.0003\lambda$ ($308.1 \times 70.6 \times 1$ mm$^3$) $308.1 \times 70.6 \times 1$ mm$^3$ and operates at 85 MHz. The bandwidth is 1.96% (84.2~85.87 MHz) based on VSWR 3.5:1, and has a realized gain of $-22.6$ dBi in simulation. Figure 6 shows the normalized radiation pattern of the 7-cell antenna, which confirms that the proposed antenna has an omnidirectional radiation pattern at the zeroth-order resonant frequency. Therefore, the proposed antenna may be used as an antenna for receiving military antennas [26,27].

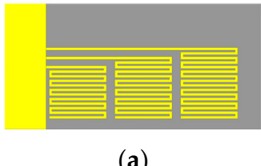

(a)

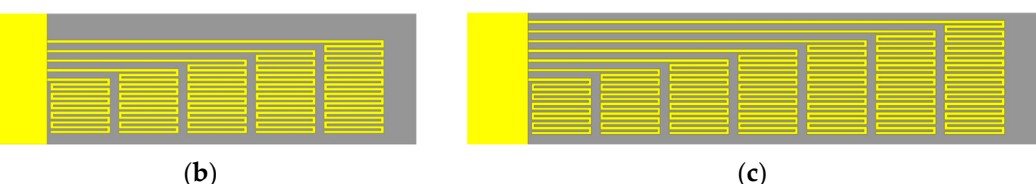

(b)

(c)

**Figure 5.** The CRLH TL antenna structure at the bottom: (**a**) 3-unit cell; (**b**) 5-unit cell; (**c**) 7-unit cell.

**Table 2.** Comparison of the antenna characteristic according to the number of CRLH unit cells.

| Number of CRLH Unit Cells | Antenna Size | BW (VSWR 3.5:1) | Antenna Gain |
|---|---|---|---|
| 3-cell | $0.047\lambda \times 0.02\lambda \times 0.0003\lambda$ ($159.9 \times 70.6 \times 1$ mm$^3$) | 1.51% (87.64~88.97 MHz) | $-31.79$ dBi |
| 5-cell | $0.07\lambda \times 0.02\lambda \times 0.0003\lambda$ ($234 \times 70.6 \times 1$ mm$^3$) | 1.7% (85.84~87.3 MHz) | $-24.47$ dBi |
| 7-cell | $0.087\lambda \times 0.02\lambda \times 0.0003\lambda$ ($308.1 \times 70.6 \times 1$ mm$^3$) | 2% (84.2~85.9 MHz) | $-22.6$ dBi |

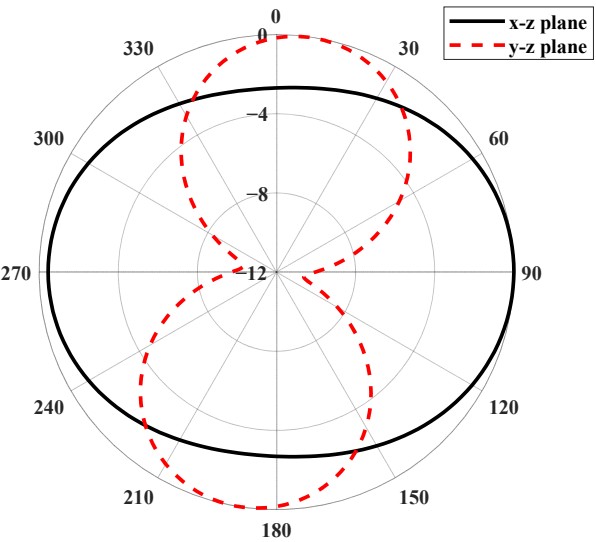

**Figure 6.** Normalized radiation pattern of the proposed CRLH TL antenna.

## 3. Fabrication and Measurement

Figure 7 shows the fabricated antenna with 7-unit cells, and Figure 8 shows the return loss of the proposed antenna. The reflection coefficient of the antenna was measured using a Rohde & Schwarz ZVA 67 Vector network analyzer [28]. There was a 1 MHz difference in resonant frequency between the fabricated antenna and the simulated antenna. This may be due to inaccuracy during the manufacturing process and experimental tolerance. The

fabricated antenna operated at 84 MHz and had a bandwidth of 2% (83.7~85.3 MHz) based on VSWR 3.5:1.

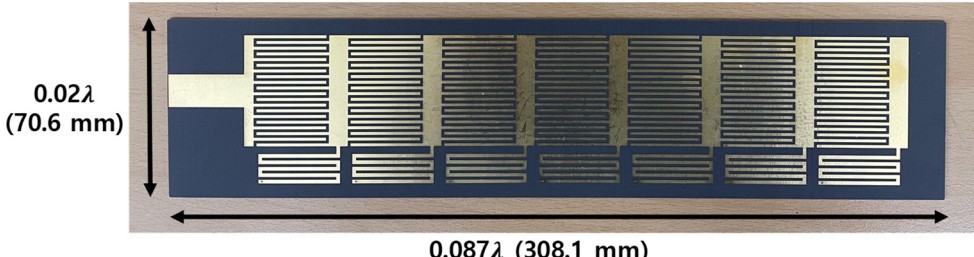

**Figure 7.** Fabricated antenna with 7-unit cells.

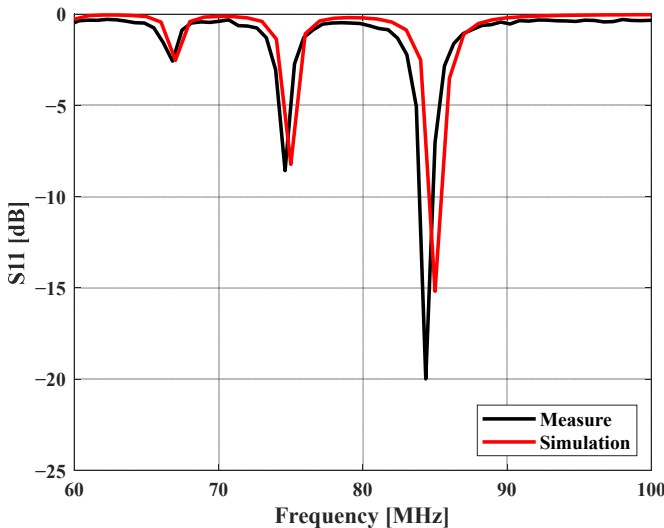

**Figure 8.** Normalized radiation pattern of the proposed CRLH antenna.

Next, to verify the performance of the proposed antenna, the received power of the antenna was simply measured and compared with simulation results. Figure 9 shows the received power measurement environment. The transmitting antenna is an OMNI-A0245 antenna from Alaris [29], and a ground plane of 1500 × 1500 mm² size was used as required in the antenna data sheet. The ground plane was wrapped in copper tape on Styrofoam. The transmitting power was generated by Agilent's E4422B ESG Series Signal Generator [30], and the received power was analyzed with Agilent's E4411B ESA-L Series Spectrum Analyzer [31]. The distance between the transmitting antenna and the receiving antenna was 30 m, and the transmitting power was set to 25 dBm and transmitted at intervals of 1 MHz. The received power of the antenna was measured by fixing the antenna in the direction of the maximum radiation ($\theta = 90°$, $\phi = 10°$). In order to compare with the simulation results, the measured received power was converted into an antenna gain, as shown in Figure 10, using the Friis transmission formula [32], shown in Equation (3). $P_r$ and $P_t$ are the received power and the transmitting power, respectively. $G_r$ and $G_t$ are the gain of receiving antenna and transmitting antenna, respectively. $L_r$ and $L_t$ represent cable loss and mismatching loss as receiving loss and transmission loss, respectively. The cable loss was measured with or without cable connection, and the mismatching loss was calculated using the s-parameter characteristics of the data sheet for the transmitting antenna. In the case of the receiving antenna, it was calculated using the s-parameter characteristics measured using the VNA (Vector network analyzer), and the considered loss value was about −7.8 dBm. $\lambda$ and R are the wavelength and the distance between the antennas, respectively. Since the proposed antenna operates at a frequency below 100 MHz ($1\lambda = 3$m), it is hardly affected by surrounding objects. In addition, since it has a highly miniaturized size compared to the wavelength, and thus satisfies the far-field condition, it

was calculated by applying the Friis transmission formula. From the calculated results, it can be seen that the proposed antenna can obtain −22.7 dBi at 84 MHz and can be used as a receiving antenna [26,27]. In the band where the antenna operates, the received power did not exceed −40 dBm, so it was possible to measure it and it could be converted into an antenna gain. However, due to the low SNR performance of the spectral analyzer used in the experiment, the signal was not detected by noise at frequencies with antenna gain of approximately −40 dBi or less. Actually, the detectable signal was about 50 dBm or more, and the expected received power in the range of 60~75 MHz was −50 dBm or less, so the received signal was not detected.

$$\frac{P_r}{P_t} = \left(\frac{\lambda}{4\pi R}\right)^2 G_t G_r L_t L_r \tag{3}$$

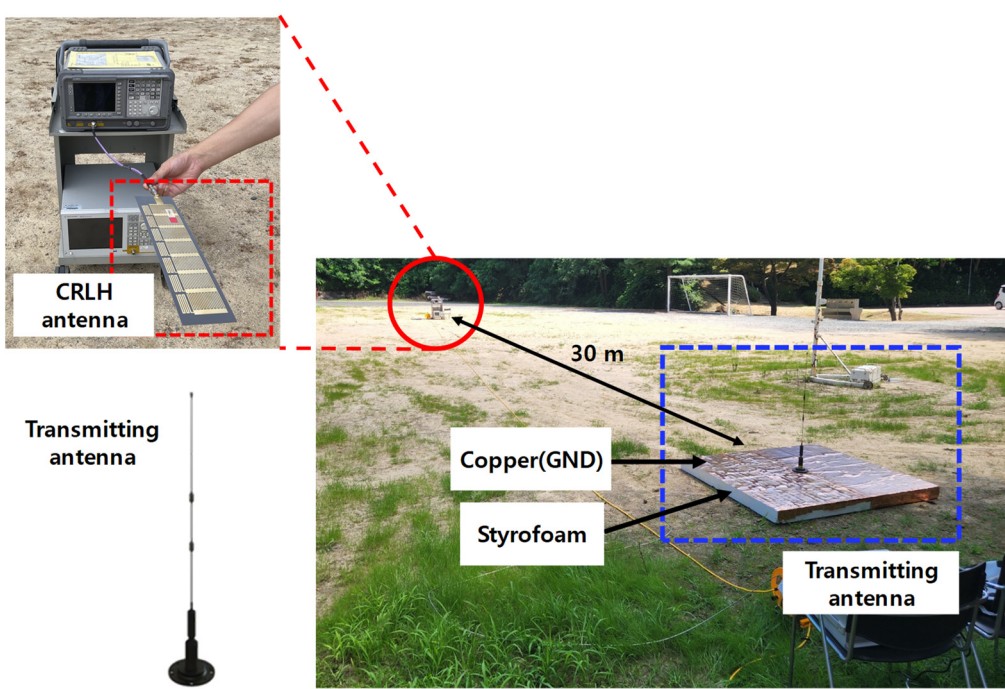

**Figure 9.** Receiving power measurement setup of the proposed antenna.

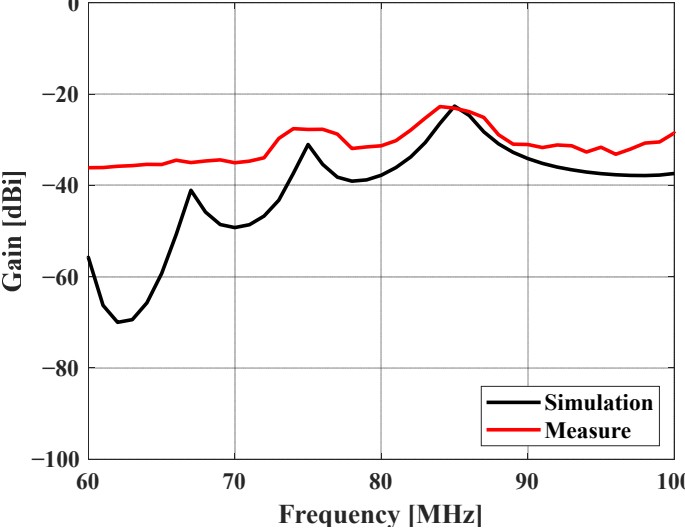

**Figure 10.** Measured and simulated gain of the proposed antenna.

Table 3 compares the commercialized military antenna and the previous CRLH transmission line antenna with the proposed antenna in this paper. The commercialized military antennas have high gain but are limited in the operation of communication equipment due to the antenna length of 1~2 m. Moreover, the CRLH transmission line antenna studied previously has an extremely low gain. However, the proposed antenna in this paper has a miniaturized antenna size in the VHF band and shows that it can be used as a receiving antenna.

**Table 3.** Comparison of the antenna performance with the previous work.

| Reference | Antenna Size | BW (VSWR 3.5:1) | Antenna Gain |
|---|---|---|---|
| [1] | 0.13λ (1300 mm) | 100% (30~90 MHz) | −10 dBi (average gain) |
| [3] | 0.234λ (2340 mm$^3$) | 98% (30~88 MHz) | 2 dBi (@ 88 MHz) |
| [15] | 0.14λ × 0.16λ × 0.01λ (24.8 × 22 × 1.5 mm$^3$) | 4.1% (1910~1990 MHz) | −6.9 dBi (@ 1950 MHz) |
| [23] | 0.021λ × 0.017λ × 0.002λ (24.8 × 22 × 1.5 mm$^3$) | 5% (395~415 MHz) | −38 dBi (@402 MHz) |
| [24] | 0.103λ × 0.043λ × 0.0008λ (191.4 × 80 × 1.52 mm$^3$) | 5% (158~166 MHz) | −26.5 dBi (@160 MHz) |
| Proposed Antenna | 0.087λ × 0.02λ × 0.0003λ (308.1 × 70.6 × 1 mm$^3$) | 2% (83.7~85.4 MHz) | −22.7 dBi (@84 MHz) |

## 4. Conclusions

In this paper, we proposed a miniaturized antenna based on a CRLH transmission line in the VHF band using meander lines at both the top and bottom. The proposed antenna was miniaturized by using the zeroth-order resonant frequency of the CRLH transmission line, and the meander lines connected to the top and bottom were used to increase the parallel inductance. The unit cell consisted of an interdigital capacitor and a meander line, and was designed to increase the length of the meander line as the number of cells increases. The size of the unit cell was 0.01λ × 0.02λ × 0.0003λ (37 × 70.6 × 1 mm$^3$). Antennas with 3-, 5-, and 7-unit cells were designed, and it was confirmed that the gain of the antenna increases as the number of unit cells increases. The optimized antenna consisted of 7-unit cells and operated at 84 MHz. The overall size of the antenna was 0.087λ × 0.02λ × 0.0003λ (308.1 × 70.6 × 1 mm$^3$). The antenna had a bandwidth of 2% (83.7~85.4 MHz) and an antenna gain of −22.7 dBi. Therefore, the proposed antenna can be used for military applications operating in the VHF band without increasing the physical size of the antenna. We will conduct a study to improve the bandwidth of the proposed antenna.

**Author Contributions:** Conceptualization, S.L. and Y.B.P.; methodology, S.L.; software, S.L.; validation, S.L.; formal analysis, S.L.; investigation, S.L.; resources, S.L.; data curation, S.L.; writing—original draft preparation, S.L.; writing—review and editing, S.L. and Y.B.P.; visualization, S.L.; supervision, Y.B.P.; project administration, Y.B.P.; funding acquisition, Y.B.P. All authors have read and agreed to the published version of the manuscript.

**Funding:** This work has been supported by the Future Combat System Network Technology Research Center program of Defense Acquisition Program Administration and Agency for Defense Development (UD190033ED).

**Institutional Review Board Statement:** Not applicable.

**Informed Consent Statement:** Not applicable.

**Data Availability Statement:** Not applicable.

**Conflicts of Interest:** The authors declare no conflict of interest.

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
