# Peer review of "A VHF Band Small CRLH Antenna Using Double-Sided Meander Lines"

_applsci, doi:10.3390/app122010676_

Round 1

Reviewer 1 Report

The paper presents an electrically small antenna operating the the VHF band for potential military applications. The proposed antenna show small gain improvement over prior designs, but with smaller bandwidth. The antenna is fabricated and tested the measurement and simulation results show good agreement.

In general, the structure of the paper is good, but the text contains many grammatically incorrect sentences. In particular the introduction needs improvement. Besides this, I have the following comments:

-The introduction should be reviewed as it contains many grammatically incorrect sentences. In particular the motivation outlined in lines 38-49 could be much clearer.

- The symbols in Equation (1) is not presented in the text.

- Please correct the symbols in lines 92-93.

- The dispersion diagram shown in Fig. 4 looks quite unnatural. Perhaps this is due to the poor resolution of the graph. Moreover, what is the purpose of this figure in terms of the antenna design? This is not clear in the text.

- Not all sub figures in Fig. 5 are needed. They are highly repetitive. Instead keep the 7-unit cell and mark the other editions with fewer units on that.

- The antenna gain is not the same in all directions, which can be seen in Fig. 7. Therefore, it is important to specify in which direction the gain is given, when a single value presented, e.g. in Table 1. It is also ok to write 'maximum' gain, but somewhere it needs to be specified in which direction that is.

- Are you sure that Friis law is applicable in this configuration i.e. the distance between the antenna are in the far-field? This depends on the size of the antennas and surrounding objects (such as the ground plane). Furthermore, the gain of the antennas are also affected by nearby objects such as the large ground plane or the chair, the receiving antenna is put on. Have this been taking into account?

- At last, I would like the Authors to comment of the small bandwidth on the antenna. Is this bandwidth acceptable for the specified military application? Furthermore, is there any way to improve this without increasing the size of the antenna?

Reviewer 2 Report

(1) Applying Composite Right/Left Handed(CRLH) transmission line and meander lines to the design of VHF antennas has been usual approaches. The additional meander lines at both top and bottom of the antenna structure would not be considered as some significant innovation.

(2) Logically, the operating bandwidth of the antenna with 5-unit cell should be between the operating bandwidths of the antennas with 3- and 7-unit cell. However, Table 1 shows that the operating bandwidth of the antenna with 5-unit cell is above that of the antenna with 3-unit cell. Please give detailed explanation and physical interpretation for this illogical result.

(3) Please give convincing detailed explanation for the reason why the difference between the measured and simulated gain of the proposed antenna is so large in the range of 60 – 75 MHz in Figure 11.

(4) The title of Table 2 is missing.

(5) There are some errors in grammar and expressions, for instance, on page 2, line 68 -69, “it resonates at the frequency which the impedance imaginary part becomes zero” should be it resonates at the frequency where the impedance imaginary part becomes zero”.

Round 2

Reviewer 2 Report

(1) The authors have properly addressed the comments and concerns of this reviewer and revised their manuscript accordingly.

(2) There are still some errors in grammar and expressions, for instance, on page 1, line 21 - 23, “The antenna for military applications mainly use dipole, monopole, and whipped antennas operating in the HF/VHF/UHF band, and require omnidirectional radiation patterns due to limited communication environments [1-3]” should be The antennas for military applications mainly use dipole, monopole, and whipped antennas operating in the HF/VHF/UHF band, and require omnidirectional radiation patterns due to limited communication environments [1-3]”.

The authors need to carefully check the whole manuscript and correct all the errors in grammar and expressions and present the contents smoothly.
